# XMask3D: Cross-modal Mask Reasoning for Open Vocabulary 3D Semantic Segmentation

**Ziyi Wang**[*]  **Yanbo Wang**[*]  **Xumin Yu**  **Jie Zhou**  **Jiwen Lu**[†]

Department of Automation, Tsinghua University, China

{wziyi22, wyb23, yuxm20}@mails.tsinghua.edu.cn;
{jzhou, lujiwen}@tsinghua.edu.cn

## Abstract

Existing methodologies in open vocabulary 3D semantic segmentation primarily concentrate on establishing a unified feature space encompassing 3D, 2D, and textual modalities. Nevertheless, traditional techniques such as global feature alignment or vision-language model distillation tend to impose only approximate correspondence, struggling notably with delineating fine-grained segmentation boundaries. To address this gap, we propose a more meticulous mask-level alignment between 3D features and the 2D-text embedding space through a cross-modal mask reasoning framework, **XMask3D**. In our approach, we developed a mask generator based on the denoising UNet from a pre-trained diffusion model, leveraging its capability for precise textual control over dense pixel representations and enhancing the open-world adaptability of the generated masks. We further integrate 3D global features as implicit conditions into the pre-trained 2D denoising UNet, enabling the generation of segmentation masks with additional 3D geometry awareness. Subsequently, the generated 2D masks are employed to align mask-level 3D representations with the vision-language feature space, thereby augmenting the open vocabulary capability of 3D geometry embeddings. Finally, we fuse complementary 2D and 3D mask features, resulting in competitive performance across multiple benchmarks for 3D open vocabulary semantic segmentation. Code is available at `https://github.com/wangzy22/XMask3D`.

## 1   Introduction

As the integration of vision and language in deep learning continues to expand, text descriptions are increasingly utilized in visual generation [37, 50, 39, 35, 26, 40] and perception [36, 22, 23, 13, 45, 12] tasks. This integration enhances the adaptability of models in real-world applications and improves user experiences in customized artificial intelligence systems. Open vocabulary 3D semantic segmentation exemplifies a perception task that is trained on *base* categories and demands robust extrapolation capabilities to discriminate fine-grained geometry in *novel* categories that are invisible during training. The base and novel classes are only linked by the shared open vocabulary within the language space. However, constructing a shared 3D-text space while maintaining precise, modality-specific representation remains a significant challenge. Successfully addressing this issue would advance open vocabulary 3D semantic segmentation, facilitating virtual reality interactions and manipulations, and thereby contributing to the development of user-friendly robotics and autonomous driving technologies.

Existing approaches for open vocabulary 3D perception predominantly aim to bridge the gap between 3D and text representations by using the 2D modality as an intermediary. One line of research [34,

---

[*]Equal contribution.  [†]Corresponding author.

38th Conference on Neural Information Processing Systems (NeurIPS 2024).

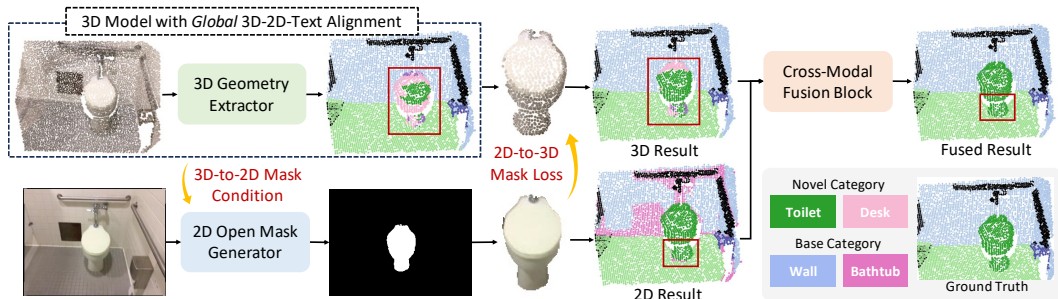

Figure 1: **The overall framework of XMask3D.** The 3D model with only *coarse* 3D-2D-text alignment struggles to segment novel categories with accurate boundaries. We propose to incorporate a 2D open mask generator conditioned on global 3D geometry features to create geometry-aware segmentation masks of novel categories. Then we apply fine-grained mask-level regularization on 3D features, thereby enhancing the dense open vocabulary capability of the 3D model. The cross-modal fusion block leverages the strengths of both branches to achieve optimal results.

11, 47, 16] suggests aligning 3D embeddings with vision-language embedding spaces through global, patch-wise, or point-wise contrastive loss, while another line [49, 28, 44] investigates distilling open vocabulary knowledge from foundational vision-language models into 3D models. However, global feature alignment and model distillation techniques tend to overlook fine-grained 3D geometric details and produce coarse results. Conversely, point-wise contrastive learning is prone to noise and outliers. Although patch-wise feature alignment offers a compromise, a single patch in 2D may correspond to multiple unrelated and discontinuous regions in 3D, which can be misleading and ineffective.

In this paper, we propose a more precise and consistent mask-level alignment between 3D features and the 2D-text embedding space, achieved through our proposed cross-modal mask reasoning method. The proposed XMask3D model comprises a 3D branch, a 2D branch, and a fusion block. The 3D branch, adaptable as any point cloud segmentation model, excels in capturing geometric features but struggles with novel category extrapolation. Conversely, the 2D branch serves as a mask generator, which predicts masks with embeddings aligned to the vision-language feature space but lacks 3D spatial perception capabilities. The fusion block allows these two branches to complement each other. Specifically, we propose utilizing the denoising UNet of a pre-trained text-to-image diffusion model with advanced vision-language modeling capabilities as the 2D mask generator. Diffusion model's exceptional control over text-driven image generation demonstrates strong potential for creating fine-grained segmentation masks of novel categories. To promote thorough interaction between the two modality branches, we propose three mask-level techniques: (1) **3D-to-2D Mask Generation.** We condition the denoising mask generator on global point cloud features, producing geometry-aware masks that are better suited for transferring to the 3D modality. (2) **2D-to-3D Mask Regularization.** We apply mask-level regularization on 3D features to align with the vision-language embedding space. This enhances the open vocabulary capability of 3D features for novel categories while preserving fine-grained geometric information. (3) **3D-2D Mask Feature Fusion.** We merge mask features from both modalities in the fusion block, enhancing the synergy between 2D and 3D features.

We conduct extensive experiments on multiple benchmarks of different datasets, including Scan-Net20 [9], ScanNet200 [38], and S3DIS [1] datasets, to evaluate the effectiveness of our proposed method. XMask3D demonstrates competitive performance across all benchmarks. Additionally, we perform thorough ablation studies and provide intuitive visualizations to showcase the contribution of each proposed mask-level technique. In conclusion, the contributions of this paper can be summarized as follows:

- We propose a novel XMask3D framework that, for the first time, leverages the denoising UNet of a generative text-to-image diffusion model for open vocabulary 3D perception.
- We introduce 3D-to-2D mask generation, 2D-to-3D mask regularization, and 3D-2D mask feature fusion techniques to enhance meticulous mask-level 3D-2D-text feature alignment and strengthen cross-modal feature synergy.
- We demonstrate the effectiveness of XMask3D on multiple benchmarks of various datasets and show outstanding performance. Ablation studies further convince the contribution of each proposed mask-level technique.

## 2 Related Work

### 2.1 2D Open Vocabulary Segmentation

Open vocabulary perception is a recently emerged research problem that focuses on enabling perception models to recognize novel categories that are invisible during supervised training, relying solely on a shared language vocabulary with the base categories. The key to addressing this problem lies in the Vision-Language model, which creates a shared embedding space for images and texts. Based on the types of Vision-Language models, previous literature on 2D open vocabulary segmentation can be broadly divided into two approaches: utilizing Vision-Language perception models like CLIP [36] or leveraging Vision-Language generation models like the diffusion model [18, 37].

Although traditional Vision-Language perception models like CLIP are primarily designed for classification, open vocabulary segmentation can also be viewed as a dense classification task. Consequently, several studies have proposed aligning dense image features with 2D-text embeddings, a concept pioneered by LSeg [23]. Successor models have introduced various techniques to enhance the feature alignment process, including attention-based combinations [30, 12], mask embedding decoupling [10], side network injection [46, 24], and cross-modal aggregation [6].

Generative-based methods utilize Vision-Language generation models, such as diffusion models, to produce segmentation masks that can be extrapolated to open vocabulary categories. Since the diffusion model can generate semantically meaningful images based on text conditions, its intermediate features effectively represent the vision-language embedding space. ODISE [45] first proposed using the intermediate features from the denoising UNet of a pre-trained diffusion model as input to a mask generator for segmentation. Other works [25, 21, 41] leverage the strong generative capabilities of the diffusion model to create prototypes or augmented image-mask pairs, thereby enhancing the open vocabulary capacity of the segmentation model from a data perspective.

### 2.2 3D Open Vocabulary Segmentation

In 3D vision, Semantic Abstraction [15] opens up the avenue to leverage Vision-Language models for open-world 3D scene understanding. Subsequent studies have primarily developed two types of methods to address 3D open vocabulary segmentation: feature alignment and model distillation.

The principle of feature alignment methods is to explicitly pull 3D representations towards the vision-language embedding space, using the 2D modality as a mediator to establish 3D-text relationships. OpenScene [34] employs a cosine similarity loss between point cloud features and image CLIP features, integrating them for open vocabulary perception. PLA [11] introduces hierarchical 3D caption pairs to progressively align scene-level, view-level, and entity-level features with the CLIP feature space in a coarse-to-fine manner. Its successor, RegionPLC [47], further introduces region-level captions with sliding windows and object bounding boxes, while CLIP-FO3D [49] similarly divides super-pixels for finer feature alignment. UniM-OV3D [16] utilizes a pre-trained point-text model, PointBind [14], to enforce uni-modality representation learning of point clouds, images, depth maps, and texts. OV3D [20] proposes leveraging foundation models to establish point-entityText associations through pixel, thereby enhancing open-vocabulary recognition within the 3D domain.

Model distillation methods typically involve selecting a foundational Vision-Language model and transferring its knowledge to a 3D network using paired point clouds and image data. Seal [28] introduces spatial contrastive learning and temporal consistency regularization to distill vision foundation models for point cloud sequence segmentation in an open vocabulary setting. 3D-OVS [27] aims to distill CLIP [36] and DINO [2, 33] into a neural radiance field [32] using novel alignment losses for 3D perception. Xiao et al. [44] propose object-level and voxel-level distillation losses for fine-grained 3D open vocabulary panoptic segmentation.

## 3 Approach

### 3.1 Overview

The detailed architecture of XMask3D is depicted in Figure 2. It consists of three components: a 3D geometry extraction branch, a 2D mask generation branch, and a 3D-2D feature fusion module. The 3D geometry extraction branch is an encoder-decoder segmentation network, which can be

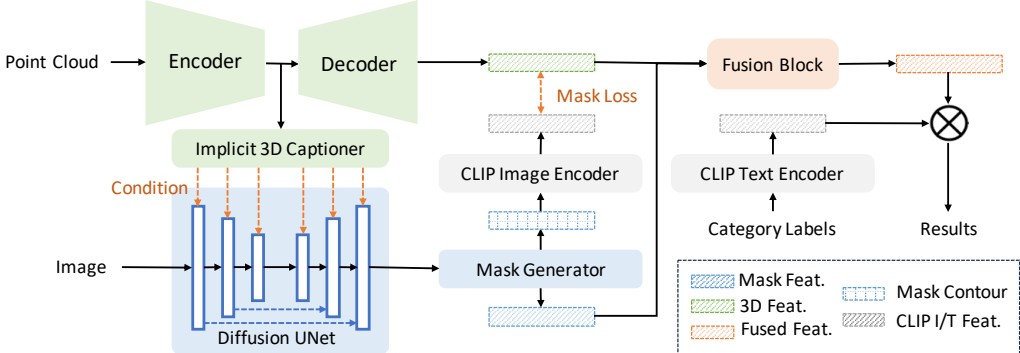

Figure 2: **The detailed architecture of XMask3D.** We introduce an auxiliary 2D branch, which utilizes global point cloud features as conditional input to generate open vocabulary masks. The contour of the mask is utilized for regularization at the mask level on 3D features, and the embeddings of the mask are fused with the 3D features to enhance cross-modal complementarity.

implemented using any off-the-shelf architectures such as sparse convolution networks [7, 8] or Transformer-based networks [51, 43, 42]. These frameworks are specifically designed to extract geometric features $F_{3d} \in \mathbb{R}^{N \times C}$ from 3D point clouds, where $N$ and $C$ represent the numbers of points and feature dimensions, respectively. However, $F_{3d}$ is discriminative only on base categories trained with supervision, relying on a pre-defined classifier implemented as a fully connected layer or a multi-layer perceptron. Therefore, it's essential for $F_{3d}$ to be aligned with the vision-language feature space to enable unbounded open vocabulary segmentation, which is performed via per-point similarity comparisons between $F_{3d}$ and CLIP [36] text embeddings of given category names: $F_{text} = \mathcal{E}_{CLIP_T}(\mathbf{C}_{name}) \in \mathbb{R}^{L \times C}$, where $L$ represents the number of categories, $\mathcal{E}_{CLIP_T}$ denotes the CLIP text encoder, and $\mathbf{C}_{name}$ stands for the category name such as *Table, Chair*.

Previous literature on feature alignment techniques varies from global or point-level contrastive learning to vision-language model distillation. However, none of these methods simultaneously achieves fine-grained precision and robustness to outliers. To address this limitation, we introduce a cross-modal mask reasoning method that performs mask-level feature alignment. Specifically, we propose a 2D mask generation branch in conjunction with the 3D branch to generate 2D masks with open vocabulary capability and use these masks to regularize $F_{3d}$. Detailed information on mask generation and mask regularization can be found in Section 3.2 and Section 3.3, respectively.

From the 2D mask generator, we obtain the 2D mask embeddings $G_{2d} \in \mathbb{R}^{M \times C}$ and binary mask maps $\mathcal{M}_{2d} \in \mathbb{R}^{H \times W \times M}$, where $M$ represents the number of candidate masks, and $H$ and $W$ denote the height and width of the input images. Using the camera intrinsic matrix $K \in \mathbb{R}^{3 \times 3}$ and the view projection matrix $V \in \mathbb{R}^{4 \times 4}$, we can establish associations between pixels in image $I$ and surface points in the corresponding point cloud $P$. This allows us to derive back-projected 3D binary masks $\mathcal{M}_{3d} \in \mathbb{R}^{N' \times M}$ from $\mathcal{M}_{2d}$. Here, $N'$ signifies the number of points that correspond to image pixels, following the relationship $N' < N, N' < H \times W$. Subsequently, we derive a pseudo mask feature $F_{2d} = \mathcal{M}_{3d} \cdot G_{2d}, F_{2d} \in \mathbb{R}^{N' \times C}$, with the subscript 2d indicating its origin from the 2D branch.

Having obtained $F_{3d}$ from the 3D geometry extraction branch and $F_{2d}$ from the 2D mask generation branch, we implement a 3D-2D fusion block to combine these cross-modal and complementary features, resulting in $F_{fuse} \in \mathbb{R}^{N \times C}$. Detailed information on this process can be found in Section 3.4. The final open vocabulary 3D semantic segmentation output $O \in \mathbb{R}^{N \times L}$ is then calculated via:

$$O = \operatorname{argmax}_N \frac{F_{fuse} \cdot F_{text}^T}{\|F_{fuse}\|\|F_{text}\|} \tag{1}$$

Detailed training objectives can be found in Section 3.5.

### 3.2 3D-to-2D Mask Generation

**Design Insights.** An optimal 2D branch is expected to exhibit robust open vocabulary capabilities, enabling it to predict accurate masks for novel categories. To this end, we employ the denoising UNet of the renowned text-to-image diffusion model [37] to extract features from the well-established text-2D embedding space, followed by a mask generator to convert features into segmentation masks.

We favor the generative diffusion model over the commonly adopted CLIP [36] model for two primary reasons. Indicated by Prompt-to-Prompt [17], the cross-attention maps from the intermediate layers of the diffusion model exhibit a high correlation with text concepts. Consequently, a well-trained diffusion model constructs a superior vision-language feature space, which can be effectively leveraged for open vocabulary perception. Moreover, the diffusion model provides precise text control over dense pixel generation, demonstrating a higher potential for generating fine-grained segmentation masks compared to the CLIP model, which relies on coarse global feature alignment.

**Preliminary: Open Vocabulary Mask Generation with diffusion.** The inference process of a text-to-image diffusion model [37] begins with a random Gaussian noise and a conditional text embedding, generating high-quality images through iterative denoising steps. ODISE [45] was the first to propose the use of the denoising UNet of a pre-trained diffusion model for open vocabulary *2D* segmentation. Given an input image $x$, a noisy image $x_t$ is first sampled at time step $t$:

$$x_t = \sqrt{\bar{\alpha}_t}x + \sqrt{1 - \bar{\alpha}_t}\epsilon, \quad \epsilon \sim \mathcal{N}(0, \mathbf{I}) \tag{2}$$

where $\bar{\alpha}_t = \prod_{k=1}^{t} \alpha_k$, and $\alpha_1, \ldots, \alpha_T$ are pre-defined noise schedule. Then the diffusion model's visual representation $f$ can be computed via the denoising step:

$$f = \text{UNet}(x_t, \text{MLP} \circ \mathcal{V}(x)) \tag{3}$$

where the denoising UNet is the building block of the diffusion model, MLP stands for multi-layer perceptron, and $\mathcal{V}$ is a frozen CLIP image encoder to encode $x$ into the vision-language embedding space. $\mathcal{V}(x)$ is the implicit caption embedding which serves as the text condition input to the diffusion model. Subsequently, the mask generator, implemented with Mask2Former [4, 3], uses the feature $f$ to produce $M$ class-agnostic binary masks $\mathcal{M}_{2\text{d}}$ and their corresponding mask embedding features $F_{\text{mask}}$. Since $f$ is highly representative of the vision-language feature space, the model is inherently capable of generating open vocabulary segmentation masks and embeddings. The experimental results of ODISE strongly confirm this hypothesis.

**Geometry-aware Mask Generation.** In XMask3D, the 2D mask generation branch is implemented by a variant of the ODISE model. We propose an *Implicit 3D Captioner* that takes the global 3D feature $f_{3\text{d}} \in \mathbb{R}^{1 \times C_g}$ from the 3D encoder as input, and predicts the implicit condition embedding to be injected into the diffusion model. Then Equation 3 can be replaced by:

$$f = \text{UNet}(x_t, \text{MLP} \circ f_{3\text{d}}), \quad f_{3\text{d}} = \mathcal{E}(P) \tag{4}$$

where $\mathcal{E}$ represents the encoder of the point cloud segmentation model, and $C_g$ denotes the feature dimension of the global point cloud feature. The rationale behind this design is twofold. First, since $\text{MLP} \circ f_{3\text{d}}$ serves as the text condition for the pre-trained denoising UNet with frozen weights, the training objective of the 2D mask generation branch implicitly pushes $f_{3\text{d}}$ closer to the text-2D feature space. If $f_{3d}$ does not align with this space, the pre-trained denoising UNet will not recognize the condition, resulting in a high loss of the 2D branch. Through gradient descent, the point cloud encoder gradually distills some vision-language knowledge from the pre-trained and frozen denoising UNet. Second, $f_{3\text{d}}$ encapsulates rich 3D geometric information that the 2D branch lacks due to occlusion and dimensional compression issues inherent in images. Using $f_{3\text{d}}$ as the condition for the 2D branch encourages the model to produce geometry-aware mask outlines and embeddings, facilitating the back-projection of 2D masks into 3D space. The effectiveness of the proposed geometry-aware mask generation will be validated through quantitative ablation comparisons in Section 4.3.

### 3.3 2D-to-3D Mask Regularization

Although some vision-language knowledge from the diffusion model is distilled to the point cloud encoder $\mathcal{E}$ via the proposed Implicit 3D Captioner, the 3D feature $f_{3\text{d}}$ still deviates from the 2D-text embedding space. This is because there is no constraint on the point cloud decoder $\mathcal{D}$, and the encoder distillation via gradient descent is inherently weak. Consequently, it is crucial to introduce contrastive regularization in the training pipeline to explicitly align 3D features with the shared 2D-text embedding space. Existing contrastive learning methods [34, 11] between 3D features and 2D-text features typically explore global, patch-wise, or point-wise relations. However, global contrastive learning is too coarse, and point-wise feature alignment is prone to noise. While patch-wise contrastive learning is more fine-grained and robust, it still lacks semantic clarity. A patch in a 2D image may correspond to multiple irrelevant and discontinuous regions in a 3D point cloud due to depth compression, resulting in ambiguous and less representative local features in 3D.

To this end, we propose an explicit 2D-to-3D mask regularization term for fine-grained and consistent feature space alignment between the 3D and 2D-text modalities. Specifically, we extract 3D mask embeddings $G_{3\text{d}} \in \mathbb{R}^{M \times C}$ from 3D features $F_{3\text{d}}$ using the back-projected 3D binary mask $\mathcal{M}_{3\text{d}}$:

$$G_{3\text{d}}^i = \text{avgpool}(\tilde{F}_{3\text{d}}^i), \quad 1 \leq i \leq M \tag{5}$$

where $G_{3\text{d}}^i \in \mathbb{R}^{1 \times C}$ represents the $i^{\text{th}}$ mask embedding, and avgpool signifies the average pooling operation. $\tilde{F}_{3d}^i$ is sampled from $F_{3d}$ at indices where $\mathcal{M}_{3\text{d}}^i = 1$, with $\mathcal{M}_{3\text{d}}^i$ being the $i^{\text{th}}$ binary mask.

Given the 2D binary mask $\mathcal{M}_{2\text{d}}$ and the input image $I$, we can also derive a ground truth mask CLIP feature $G_{\text{CLIP}}$ via a pre-trained CLIP model [36]. For detailed information on obtaining the ground truth $G_{\text{CLIP}}$, please refer to MaskCLIP [12] or Section A.1. Subsequently, the 2D-to-3D regularization term can be computed using a classical cosine contrastive loss:

$$\mathcal{L}_{\text{mask}} = \frac{1}{M} \sum_{i=1}^{M} \left( 1 - \frac{G_{3\text{d}}^i \cdot (G_{\text{CLIP}}^i)^T}{\|G_{3\text{d}}^i\| \|G_{\text{CLIP}}^i\|} \right) \tag{6}$$

As each mask region ideally corresponds to a distinct category, the pooled mask embedding achieves semantic consistency and representativeness. Consequently, contrastive learning at the mask level offers finer granularity than global contrast, greater robustness than point-wise contrast, and clearer distinction than patch-wise contrast. Through our proposed 2D-to-3D mask regularization, the 3D features are explicitly aligned with the 2D-text feature space, enhancing the performance of the 3D branch in open vocabulary segmentation. This progress is further substantiated by the ablation studies outlined in Section 4.3.

### 3.4 3D-2D Mask Feature Fusion

The 3D-2D mask feature fusion block is devised to merge 3D features $F_{3\text{d}} \in \mathbb{R}^{N \times C}$ with the pseudo mask feature $F_{2\text{d}} \in \mathbb{R}^{N' \times C}$ derived from the 2D branch. It is noteworthy that each element in $F_{3\text{d}}$ possesses unique and distinguishing embeddings, whereas elements in $F_{2\text{d}}$ pertaining to the same mask share identical mask embeddings. Consequently, $F_{3\text{d}}$ offers detailed geometric structural information, while $F_{2\text{d}}$ provides semantic features with robust open vocabulary capabilities. Our approach combines features from these two modalities to leverage their complementary insights, resulting in $F_{\text{fuse}}$ which excels in both precise geometry delineation and expansive semantic extrapolation. Concretely, given that $N' < N$, we selectively merge $F_{2\text{d}}$ and $F_{3\text{d}}$ solely on the $N'$ points where correspondences exist:

$$F_{\text{fuse}} = \begin{cases} \text{MLP} \circ \text{cat}(F_{3\text{d}}, F_{2\text{d}}) & \text{have correspondence} \\ F_{3\text{d}} & \text{no correspondence} \end{cases} \tag{7}$$

where cat represents concatenation. Ablation studies in Section 4.3 and visualization results in Section 4.2 will demonstrate that $F_{\text{fuse}}$ effectively integrates the strengths of both $F_{3\text{d}}$ and $F_{2\text{d}}$.

### 3.5 Training Objectives

In XMask3D, our training strategy encompasses supervised segmentation loss ($\mathcal{L}_{\text{seg}}$) computed from 3D ($\mathcal{L}_{\text{seg}}^{3\text{d}}$), 2D ($\mathcal{L}_{\text{seg}}^{2\text{d}}$), and fusion ($\mathcal{L}_{\text{seg}}^{\text{fuse}}$) modalities. We employ Cross Entropy loss for 3D and fusion segments, and for 2D, we adopt multi-head losses including Cross Entropy, Dice, and Focal Loss, following ODISE [45] and Mask2Former [4, 3] guidelines. Additionally, we follow PLA [11] to introduce a binary head and view-level contrastive loss. The binary head is optimized with Binary Cross Entropy loss ($\mathcal{L}_{\text{bi}}$) to differentiate between base and novel categories. The view-level contrastive loss ($\mathcal{L}_{\text{view}}$) is calculated between the view global feature and text embedding of the view image caption, weighted by respective coefficients ($\omega_{\text{view}}^{3\text{d}}, \omega_{\text{view}}^{2\text{d}}, \omega_{\text{view}}^{\text{fuse}}$). Detailed information about the binary head and the view-level regularization can be found in the PLA paper or in Section A.3 and A.2. In conclusion, the overall training objective can be adjusted by:

$$\mathcal{L} = \omega_{\text{seg}} \mathcal{L}_{\text{seg}} + \omega_{\text{mask}} \mathcal{L}_{\text{mask}} + \mathcal{L}_{\text{view}} + \omega_{\text{bi}} \mathcal{L}_{\text{bi}} \tag{8}$$

where $\omega_{\text{seg}}, \omega_{\text{mask}}, \omega_{\text{view}}^{3\text{d}}, \omega_{\text{view}}^{2\text{d}}, \omega_{\text{view}}^{\text{fuse}}, \omega_{\text{bi}}$, are loss weight hyperparameters.

Table 1: **Results for open-vocabulary 3D semantic segmentation on ScanNet dataset.** We evaluate the performance with hIoU, base and novel mIoU on five benchmarks with different category splits.

| Method | Scannet | | | | | | | | | ScanNet200 | | | | | |
|---|---|---|---|---|---|---|---|---|---|---|---|---|---|---|---|
| | B15/N4 | | | B12/N7 | | | B10/N9 | | | B170/N30 | | | B150/N50 | | |
| | hIoU | Base | Novel | hIoU | Base | Novel | hIoU | Base | Novel | hIoU | Base | Novel | hIoU | Base | Novel |
| LSeg-3D [23] | 0.0 | 64.4 | 0.0 | 0.9 | 55.7 | 0.1 | 1.8 | 68.4 | 0.9 | 1.5 | 21.1 | 0.8 | 3.0 | 20.6 | 1.6 |
| 3DGenZ [31] | 20.6 | 56.0 | 12.6 | 19.8 | 35.5 | 13.3 | 12.0 | 63.6 | 6.6 | 2.6 | 15.8 | 1.4 | 3.3 | 14.1 | 1.9 |
| 3DTZSL [5] | 10.5 | 36.7 | 6.1 | 3.8 | 36.6 | 2.0 | 7.8 | 55.5 | 4.2 | 0.9 | 4.0 | 0.5 | 0.7 | 3.8 | 0.4 |
| PLA [11] | 65.3 | 68.3 | 62.4 | 55.3 | 69.5 | 45.9 | 53.1 | 76.2 | 40.8 | 11.4 | 20.9 | 7.8 | 10.1 | 20.9 | 6.6 |
| OpenScene [34] | 65.7 | 68.8 | 62.8 | 56.8 | 61.5 | 51.7 | 54.3 | 71.8 | 43.6 | 14.2 | 22.5 | 10.4 | 15.2 | 23.5 | 11.2 |
| OV3D [20] | 72.4 | 70.2 | 74.7 | 68.5 | 74.1 | 63.7 | 64.8 | 77.6 | 55.6 | – | – | – | – | – | – |
| XMask3D | 70.0 | 69.8 | 70.2 | 61.7 | 70.2 | 55.1 | 55.7 | 76.5 | 43.8 | 18.0 | 27.8 | 13.3 | 15.5 | 24.4 | 11.4 |

## 4 Experiments

### 4.1 Experiment Settings

**Datasets.** In accordance with prior literature, our research conducts experimentation on two prominent indoor scene datasets: ScanNet [9] and S3DIS [1]. ScanNet, a foundational dataset in this domain, comprises 1201 scenes allocated for training and 312 scenes designated for validation. Each scene within ScanNet furnishes point cloud data, multi-view images, and corresponding camera pose matrices. Similarly, S3DIS offers analogous data modalities, encompassing 271 rooms across six distinct indoor environments. Conforming to established conventions, we reserve Area 5 of S3DIS for validation purposes, ensuring consistency with prior methodologies.

**Category Partition.** In alignment with previous research, we exclude the *otherfurniture* class and partition the remaining classes into three benchmarks: B15/N4, B12/N7, and B10/N9. Here, *B15* signifies the 15 fundamental categories that remain visible and supervised during the training process, while *N4* denotes the presence of 4 novel categories introduced during evaluation. For the ScanNet variant featuring 200 classes [38], we adopt a similar approach, dividing the dataset into B170/N30 and B150/N50 benchmarks, each representing a distinct configuration of base and novel categories. Similarly, in the case of S3DIS, comprising 13 classes, we disregard the *clutter* class and organize the dataset into B8/N4 and B6/N6 benchmarks. Detailed information can be found in Section A.4.

**Metrics.** Following PLA [11], we present the mean Intersection over Union (mIoU) scores separately for both base and novel categories to assess open vocabulary segmentation performance. Additionally, to provide a comprehensive evaluation of the segmentation capability, we report the harmonic mean IoU (hIoU) derived from the mIoU scores of base and novel categories. This holistic metric offers insights into the overall segmentation efficacy across the dataset.

**Implementation Details.** Our implementation incorporates MinkUNet [7] as the 3D branch and ODISE [45] as the 2D branch within the architecture. For the vision-language model, we opt for CLIP-L [36]. The training regimen for the XMask3D model involves utilizing the AdamW optimizer [29] with a Cosine learning rate scheduler. We train the model for 150 epochs on 4 NVIDIA A800 GPUs, employing a batch size of 64. Notably, we introduce mask-level regularization to the training pipeline after the initial 50 epochs. This decision is motivated by the observation that the quality of mask prediction at the onset of training may be suboptimal, making the mask-level contrastive loss ineffective and potentially misleading. Detailed information regarding hyperparameter selections is provided in Section A.4.

### 4.2 Main Results

**Quantitative Comparisons.** From Table 1 and Table 2, our proposed XMask3D outperforms previous methods across most benchmarks, irrespective of the novel category proportion or dataset sources. The performance indicates that XMask3D is a robust and generalizable method for open vocabulary 3D semantic segmentation. Notably, we compare XMask3D with our baseline method, PLA, on novel category performance. On the ScanNet dataset, XMask3D demonstrates improvements ranging from 7.4% to 20.0% over PLA. On the ScanNet200 dataset, XMask3D surpasses PLA by an impressive

Table 2: **Open-vocabulary 3D semantic segmentation results on S3DIS dataset**. We report hIoU, base mIoU and novel mIoU metrics. Best open-vocabulary results are highlighted in **bold**.

| Method | S3DIS | | | | | |
|---|---|---|---|---|---|---|
| | B8/N4 | | | B6/N6 | | |
| | hIoU | Base | Novel | hIoU | Base | Novel |
| LSeg-3D [23] | 0.1 | 49.0 | 0.1 | 0.0 | 30.1 | 0.0 |
| 3DTZSL [5] | 8.4 | 43.1 | 4.7 | 3.5 | 28.2 | 1.9 |
| 3DGenZ [31] | 8.8 | 50.3 | 4.8 | 9.4 | 20.3 | 6.1 |
| PLA [11] | 34.6 | 59.0 | 24.5 | 38.5 | 55.5 | 29.4 |
| OpenScene [34] | 42.4 | 58.6 | 33.2 | 44.2 | **56.2** | 36.4 |
| XMask3D | **46.8** | **63.1** | **37.2** | **44.9** | 52.8 | **39.1** |

Table 3: **Ablations for XMask3D pipeline design**. We conduct experiments on the B12/N7 benchmark.

(a) Ablation for implicit condition of the diffusion model.

| Cond | Base | Novel | bed | chair | table | BKS | pic | sink | BT |
|---|---|---|---|---|---|---|---|---|---|
| Text | 69.5 | 52.7 | 72.9 | 60.6 | 36.7 | 70.0 | 14.3 | 44.6 | 70.3 |
| 2D | 69.7 | 53.6 | 70.6 | 63.3 | 40.9 | 68.4 | 12.4 | 51.6 | 67.8 |
| 3D | **70.2** | **55.1** | 72.5 | 62.7 | 37.3 | 70.6 | 18.6 | 51.2 | 73.0 |

(b) Ablation for mask regularization and fusion block.

| $\mathcal{L}_{\text{mask}}$ | Base | | | Novel | | | | |
|---|---|---|---|---|---|---|---|---|
| | 2D | 3D | Fuse | 2D | 3D | $(\Delta_{\text{3D}})$ | Fuse | $(\Delta_{\text{Fuse}})$ |
| ✗ | 40.1 | 63.9 | 70.0 | 30.9 | 14.0 | | 53.5 | |
| ✓ | 40.6 | 64.3 | 70.2 | 30.8 | 25.7 | (+11.7) | 55.1 | (+1.6) |

70.5% and 72.7%. On the S3DIS dataset, XMask3D shows boosts of 51.8% and 33.0% over PLA. Among these benchmarks, XMask3D achieves the highest improvements on the long-tail ScanNet200 dataset, primarily due to the introduction of the denoising UNet from the pre-trained diffusion model, which constructs a comprehensive text-2D embedding space with unlimited text descriptions.

It is noteworthy that the OpenScene results are derived from the implementation of UniM-OV3D [16], but we do not compare XMask3D with UniM-OV3D in the tables. This is because UniM-OV3D employs PointBIND [14] and CLIP2Point [19] with an already aligned 3D-text embedding space, whereas XMask3D only integrates the commonly used 2D-text space and introduces techniques to pull 3D features from any point cloud models towards the shared embedding space. Another outstanding concurrent method is OV3D [20], which primarily focuses on EntityText extraction and Point-EntityText association, while XMask3D concentrates on enhancing mask-level interaction between 2D and 3D modalities. Therefore, the contributions of OV3D and XMask3D are orthogonal and could complement each other in future explorations. Additionally, OV3D only provides experiment results on the ScanNet20 benchmarks, while its effectiveness on the ScanNet200 dataset that is more challenging and on the S3DIS dataset with different data distribution remains unexplored.

**Visualization Results.** In Figure 3, we present a comparative visualization of novel categories between XMask3D and previous methods [11, 34]. XMask3D demonstrates superior accuracy in category predictions, produces finer segmentation boundaries, and generates more cohesive mask regions. Notably, the missegmented region on the bookshelf in the second row of XMask3D is classified as picture, which appears reasonable given the corresponding part in the view image.

## 4.3 Ablation Studies

In this section, we comprehensively discuss the design choices of XMask3D through extensive ablation studies on the ScanNet B12/N7 benchmark. The results are presented in Table 3.

**Mask Generation Condition.** In Section 3.2, we introduce an Implicit 3D Captioner to convert global 3D features into implicit condition embeddings for the diffusion model. In Table 3a, we compare this implicit 3D condition with the vanilla text condition and implicit 2D condition. The text condition is generated by a ViT-GPT2 [48] captioning model and encoded via the frozen CLIP text encoder. The 2D condition is generated by the frozen CLIP image encoder and a learnable MLP, following the design in ODISE [45] (Equation 3). Our proposed implicit 3D condition outperforms the others in novel category segmentation, demonstrating that integrating the 3D global feature with the diffusion model produces the most compatible open vocabulary masks with the 3D branch.

**Mask Regularization.** In Section 3.3, we propose a fine-grained mask regularization term to align 3D features with the 2D-text embedding space. In the first line of Table 3b, we remove the fine-grained mask-level loss $\mathcal{L}_{\text{mask}}$ from the training pipeline. Besides the final results from the fusion block, we also report the intermediate results from the 2D and 3D branches. The inclusion of the mask loss results in a significant improvement of 11.7 in 3D performance on novel categories, demonstrating that our fine-grained mask-level regularization effectively brings 3D features closer to the 2D-text embedding space. Additionally, the performance gain on the fused output is 1.5, highlighting the positive impact of this regularization from 3D to fusion features.

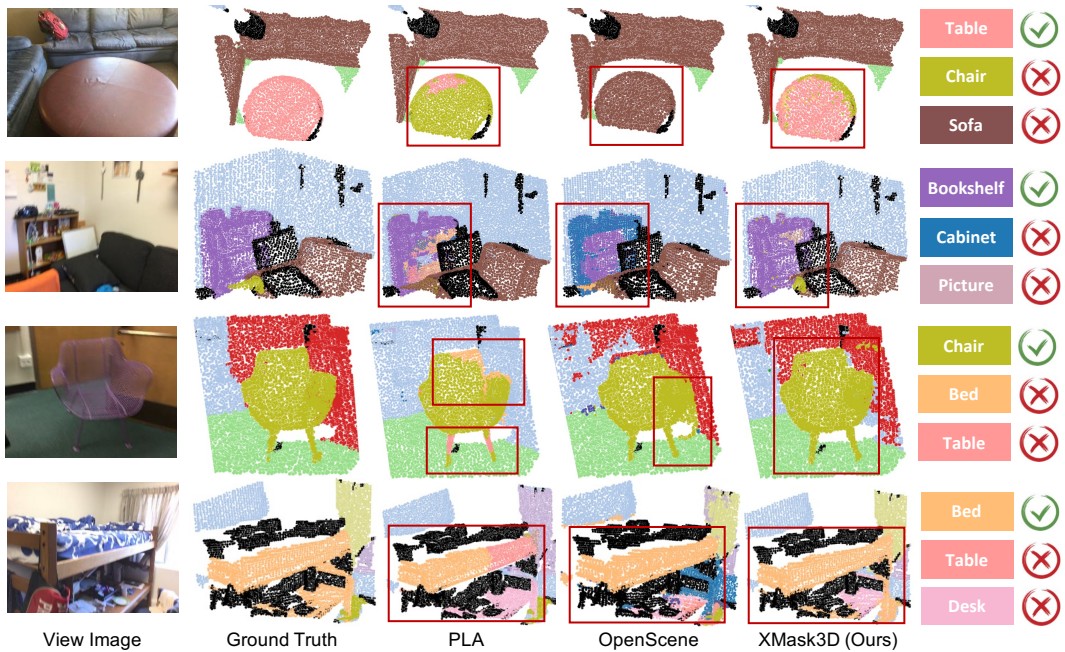

| View Image | Ground Truth | PLA | OpenScene | XMask3D (Ours) |

Figure 3: **Visualization Comparisons between XMask3D and Previous Methods.** We compare XMask3D with PLA [11] and OpenScene [34] on the novel categories table, bookshelf, chair and bed. The regions corresponding to the novel categories are highlighted in red boxes.

We also analyze the effects of mask regularization through visualizations in Figure 4. Comparing the first and second rows, the 3D segmentation results without the mask loss show inconsistency in the local region for the novel category chair, with most points misclassified as bookshelf. When the mask-level regularization is added, more points are correctly classified, resulting in a clearer segmentation mask. The visualizations of the fused outputs also demonstrate consistent enhancement in mask regularization within the final column. This is evident from the rectification of misclassified points previously labeled as table on the armrest of the right chair in the last column.

**Modality Fusion.** When comparing the 2D, 3D, and fused metrics within the same line in Table 3b, we empirically find that the 2D branch performs relatively better on novel category segmentation, while the 3D branch excels at base category segmentation. This quantitative observation aligns with our design intention: to exploit geometric knowledge via the 3D branch and to enhance the model's open vocabulary capability via the 2D branch. More importantly, the fused output outperforms both the 2D and 3D intermediate results on both base and novel splits. These results strongly support the effectiveness of our fusion design in merging the complementary knowledge from both modalities.

We also present visualization evidence regarding modal complementarity and fusion effects in Figure 4. In the first group, the 2D branch exhibits unsatisfactory results of the base category wall around the whiteboard and behind the right chair, whereas effectively segmenting the novel category chair with high quality despite minor artifacts. Conversely, the 3D branch produces an unsatisfactory mask for the chair but excels in segmenting the wall based on geometric information. The fusion block leverages the strengths of both branches, mitigates their weaknesses, and yields satisfactory outcomes. Moreover, the fusion block accurately delineates regions such as the left foot of the chair on the right, where both the 2D and 3D branches falter, highlighting its potential for integrating cross-modality knowledge. The second group displays similar results that convince the effectiveness of the proposed mask loss and cross-modality fusion design.

## 4.4 Limitations

Due to resource constraints, we only evaluate the performance of XMask3D on semantic segmentation in this study. However, the XMask3D pipeline has the potential to be extended to the instance and panoptic perception by replacing the 3D backbone with an instance or panoptic segmentation model.

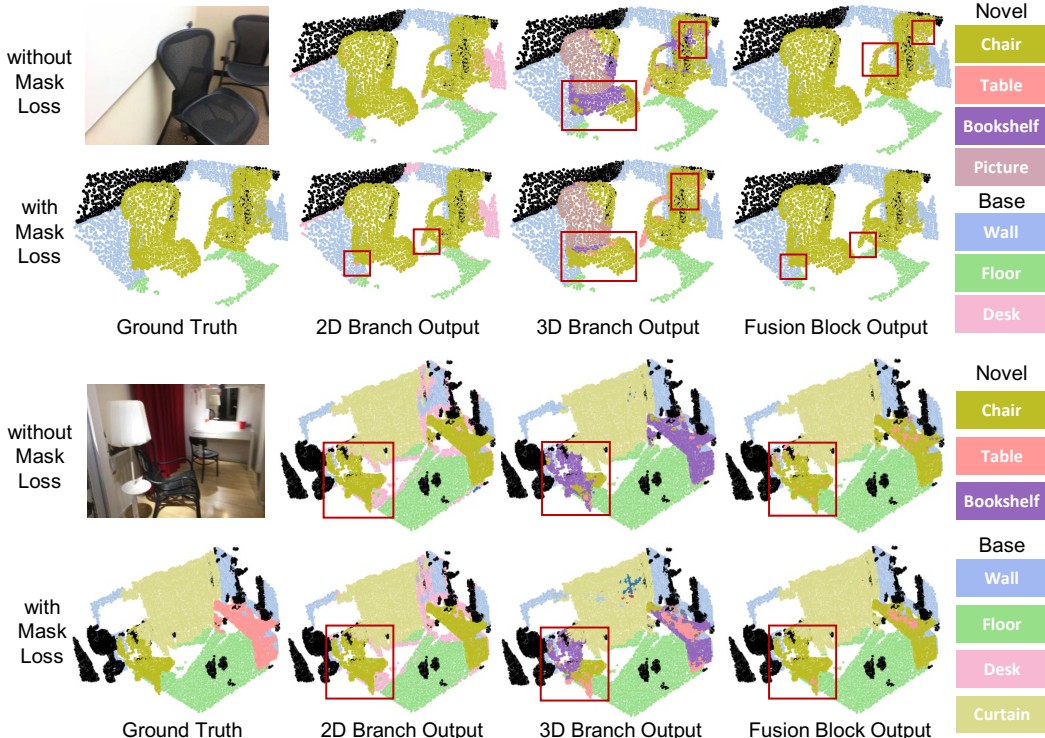

Figure 4: **Visualization Results of Ablations.** The first and second groups show results from ScanNet B15/N4 and B12/N7 benchmark, respectively. In each group, the first and second rows display segmentation results without and with the proposed mask regularization. The last three columns compare the outputs from the intermediate 2D and 3D branches with the final fusion block.

This adaptability arises from the auxiliary 2D branch, which is mask-based and capable of generating instance-level masks given appropriate annotations. Furthermore, since the XMask3D pipeline is agnostic to the 3D model used, it would be interesting to conduct experiments with more advanced point cloud models to compare performance. We hope we can exploit the strength of XMask3D with high-standard 3D backbones and a wider range of dense perception tasks in future research. Computational cost is another limitation of XMask3D, as we implement the Denoising UNet which has numerous parameters and relatively slow latency. In our future work, we plan to address this limitation by replacing the 2D branch with a more lightweight 2D open vocabulary mask generator.

## 5 Conclusion

In this paper, we present XMask3D, designed for open vocabulary 3D semantic segmentation. We propose the integration of the denoising UNet of a pre-trained diffusion model to produce geometry-aware segmentation masks conditioned on learnable implicit 3D embeddings. These binary 2D masks filter mask-level embeddings of 3D representations and apply mask regularization to enhance the open vocabulary capacity of 3D features. By fusing the 2D mask embeddings with fine-grained 3D features, we leverage the complementary knowledge from both modalities, achieving competitive performance across various benchmarks and datasets. Ablation studies and visualization comparisons further validate the effectiveness of the proposed cross-modal mask reasoning method.

## Acknowledgments and Disclosure of Funding

This work was supported in part by the National Natural Science Foundation of China under Grant 623B2063, Grant 62321005, Grant 62336004, and Grant 62125603.

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

# A  Additional Implementation Details

## A.1  Mask-level Regularization

Mask-level regularization facilitates local and finely-grained alignment between features extracted from the 3D branch of XMask3D and the 2D-text embedding space. The ground truth for mask-level loss is computed using MaskCLIP [12], utilizing the predicted segmentation mask $\mathcal{M}_{2d} \in \mathbb{R}^{M \times H \times W}$ and the view image $I \in \mathbb{R}^{3 \times H \times W}$:

$$G_{\text{CLIP}} = \text{MaskPooling}(\mathcal{V}(I), \mathcal{M}_{2d}) \tag{9}$$

where $M$ represents the number of masks, and $H$ and $W$ denote the height and width of the image, respectively. $\mathcal{V}$ refers to the pre-trained CLIP [36] image encoder.

Specifically, the view image $I$ is encoded into image tokens $T_I \in \mathbb{R}^{N \times C}$ with the pre-trained CLIP image model, where $N$ is the number of image tokens and $C$ is the dimension of the CLIP embeddings. The class token $T_C \in \mathbb{R}^{1 \times C}$ is duplicated $M$ times as the mask class tokens $T_M \in \mathbb{R}^{M \times C}$. Then $T_I, T_C, T_M$ are concatenated together to perform masked attention with frozen weights from the pre-trained CLIP image model. The attention mask is designed as

$$\mathbb{M} = \begin{bmatrix} \mathbb{F}_{(N+1) \times (N+1)} & \mathbb{T}_{(N+1) \times M} \\ \mathbb{P}_{M \times N} \quad \mathbb{F}_{M \times 1} & \mathbb{T}_{M \times M} \end{bmatrix} \tag{10}$$

where $\mathbb{T}_{m \times n}$ is an $m \times n$ True matrix, $\mathbb{F}_{m \times n}$ is an $m \times n$ False matrix and $\mathbb{P}$ is defined as:

$$\mathbb{P}_{i,j} = \begin{cases} \text{False} & \text{if mask}_i \text{ contains at least one pixel in patch}_j \\ \text{True} & \text{otherwise.} \end{cases} \tag{11}$$

where True means that this position is masked out i.e. not allowed to attend and False otherwise. Then the updated mask class tokens $T'_M$ from the masked attention layers can be regarded as the CLIP embedding of the masked regions, serving as the ground truth $G_{\text{CLIP}}$ for mask-level regularization.

## A.2  View-level Regularization

The view-level regularization facilitates coarse and high-level alignment between features extracted from the 3D branch and the 2D-text embedding space. To elaborate, upon receiving an image $I$, we initially generate its text caption using a pre-trained captioning model, ViT-GPT2 [48]. Subsequently, we employ the pre-trained CLIP [36] text encoder to encode the text caption into the 2D-text embedding space, yielding $f_{\text{view}}^T$, which serves as the ground truth for view-level regularization.

We perform average pooling operation on dense 3D point cloud features, 2D image features and fused features to obtain their global embeddings $f_{\text{view}}^{3d}, f_{\text{view}}^{2d}, f_{\text{view}}^{\text{fuse}}$. Then we implement contrastive loss between global features and the ground truth text embeddings:

$$\mathcal{L}_{\text{view}}^m = 1 - \frac{f_{\text{view}}^m \cdot (f_{\text{view}}^T)^T}{\|f_{\text{view}}^m\| \|f_{\text{view}}^T\|} \tag{12}$$

where $m = 3d, 2d, \text{fuse}$ denotes different modalities.

## A.3  Binary Head

Following PLA [11], we include a binary head to predict whether the points belong to base or novel categories. We implement a small 3D model as the binary head with minimum computation cost. The prediction $s^b$ is utilized to modulate the over-confident semantic score $s$:

$$s = s_B \cdot (1 - s^b) + s_N \cdot s^b \tag{13}$$

where $s_B$ is the semantic score computed solely on base classes with novel class scores set to zero. Similarly, $s_N$ is computed only for novel classes, setting base class scores to zero.

Table 4: **Category Partitions.** We follow PLA to split ScanNet and S3DIS into several benchmarks.

(a) ScanNet dataset.

| Partition | Base Categories | Novel Categories |
|---|---|---|
| B15/N4 | wall, floor, cabinet, bed, chair, table, door, window, picture, counter, curtain, refrigerator, showercurtain, sink, bathtub | sofa, bookshelf, desk, toilet |
| B12/N7 | wall, floor, cabinet, sofa, door, window, counter, desk, curtain, refrigerator, showercurtain, toilet | bed, chair, table, bookshelf, picture, sink, bathtub |
| B10/N9 | wall, floor, cabinet, bed, chair, sofa, table, door, window, showercurtain, curtain | bookshelf, picture, counter, desk, refrigerator, toilet, sink, bathtub |

(b) S3DIS dataset.

| Partition | Base Categories | Novel Categories |
|---|---|---|
| B8/N4 | ceiling, floor, wall, beam, column, door, chair, board | window, table, sofa, bookcase |
| B6/N6 | ceiling, wall, beam, column, chair, bookcase | floor, window, door, table, sofa, board |

Table 5: **Per-Class Results Comparison with PLA.** We compare the per-class open vocabulary segmentation results with PLA. Novel Classes are marked in blue .

(a) ScanNet dataset. *Shower c.* is short for shower curtain.

| Partition | Method | Base | Novel | wall | floor | cabinet | bed | chair | sofa | table | door | window | bookshelf | picture | counter | desk | curtain | fridge | shower c. | toilet | sink | bathtub |
|---|---|---|---|---|---|---|---|---|---|---|---|---|---|---|---|---|---|---|---|---|---|---|
| B15/N4 | PLA | 68.3 | 62.4 | 84.6 | 95.0 | 64.9 | 81.1 | 87.9 | 75.9 | 72.2 | 61.9 | 62.1 | 69.5 | 30.9 | 60.1 | 46.5 | 70.7 | 50.5 | 66.1 | 56.8 | 59.0 | 81.7 |
| | XMask3D | 69.8 | 70.2 | 84.2 | 94.7 | 69.6 | 80.8 | 86.2 | 68.4 | 74.0 | 62.1 | 60.8 | 74.4 | 29.8 | 65.3 | 52.1 | 73.2 | 57.5 | 58.9 | 86.0 | 66.5 | 83.8 |
| B12/N7 | PLA | 69.5 | 45.9 | 84.7 | 95.1 | 65.3 | 57.8 | 44.2 | 75.9 | 34.5 | 62.5 | 62.3 | 62.1 | 20.5 | 57.8 | 61.4 | 72.4 | 47.9 | 64.9 | 85.9 | 28.4 | 69.6 |
| | XMask3D | 70.2 | 55.1 | 83.3 | 94.6 | 68.6 | 72.5 | 62.7 | 76.0 | 37.3 | 62.6 | 58.0 | 70.6 | 18.6 | 64.8 | 59.6 | 71.5 | 59.1 | 60.0 | 83.9 | 51.2 | 73.0 |
| B10/N9 | PLA | 76.2 | 40.8 | 83.8 | 95.2 | 64.3 | 80.9 | 88.0 | 78.5 | 73.2 | 60.6 | 61.5 | 68.6 | 17.7 | 23.4 | 51.3 | 70.6 | 25.7 | 38.2 | 51.3 | 27.3 | 61.7 |
| | XMask3D | 76.5 | 43.8 | 83.8 | 94.7 | 67.3 | 82.6 | 89.1 | 78.8 | 72.9 | 61.2 | 62.7 | 75.2 | 17.7 | 45.9 | 54.5 | 71.9 | 28.2 | 22.9 | 59.6 | 42.3 | 47.6 |

(b) S3DIS dataset.

| Partition | Method | Base | Novel | ceiling | floor | wall | beam | column | window | door | table | chair | sofa | bookcase | board |
|---|---|---|---|---|---|---|---|---|---|---|---|---|---|---|---|
| B8/N4 | PLA | 59.0 | 24.5 | 93.9 | 97.8 | 82.9 | 0.0 | 17.2 | 15.6 | 53.7 | 35.8 | 86.3 | 05.3 | 37.3 | 43.3 |
| | XMask3D | 63.1 | 37.2 | 86.4 | 88.3 | 81.4 | 0.0 | 31.4 | 31.1 | 61.4 | 35.8 | 75.4 | 17.6 | 64.5 | 80.7 |
| B6/N6 | PLA | 55.5 | 29.4 | 93.7 | 79.1 | 80.1 | 0.1 | 28.5 | 24.1 | 08.4 | 37.6 | 87.0 | 54.0 | 24.0 | 06.9 |
| | XMask3D | 52.8 | 39.1 | 86.4 | 47.4 | 80.9 | 0.2 | 23.7 | 33.7 | 30.2 | 14.7 | 74.2 | 51.6 | 47.4 | 60.8 |

## A.4 Training and Inference Settings

**Training.** The supervised segmentation loss $\mathcal{L}_{\text{seg}}^m$ is calculated via the per-point classification Cross Entropy Loss on $N$ points:

$$\mathcal{L}_{\text{seg}}^m = \frac{1}{N} \sum_i^N \text{CrossEntropy}(\mathbf{p}, y_i) \tag{14}$$

$$\mathbf{p} = \text{Softmax}(\bar{F}_m \cdot \bar{F}_{\text{text}}^T / \tau) \tag{15}$$

where $m = 3\text{d}, 2\text{D}, \text{fuse}$ denotes different modalities, $y_i$ is the ground truth for base categories, $\bar{F}_m$ is the normalized feature, and $\tau$ is a learnable temperature parameter.

**Inference.** We follow ODISE [45] to combine the predicted logits $\mathbf{p}$ with the prediction from a text-image discriminative model to enhance the open vocabulary classification capacity of the model. Specifically, we leverage the mask-level regularization ground truth feature $G_{\text{CLIP}}$ from Section A.1 to modulate the segmentation logits:

$$\mathbf{p}_{\text{final}} \propto \mathbf{p}^\lambda \mathbf{p}_{\text{aux}}^{(1-\lambda)} \tag{16}$$

$$\mathbf{p}_{\text{aux}} = \text{Softmax}(\bar{G}_{\text{CLIP}} \cdot \bar{F}_{\text{text}}^T / \tau) \tag{17}$$

where $\bar{G}_{\text{CLIP}}$ is the normalized feature of $G_{\text{CLIP}}$, $\lambda \in [0, 1]$ is the fixed balancing factor.

**Hyper-parameters.** For all benchmarks, we set the same $\omega_{\text{seg}} = 4, \omega_{\text{view}}^{\text{3d}} = 1, \omega_{\text{view}}^{\text{2d}} = 4, \omega_{\text{view}}^{\text{fuse}} = 1.5$ as the hyper-parameter choices. The $\omega_{\text{mask}}$ and $\omega_{\text{bi}}$ are set differently across benchmarks. We

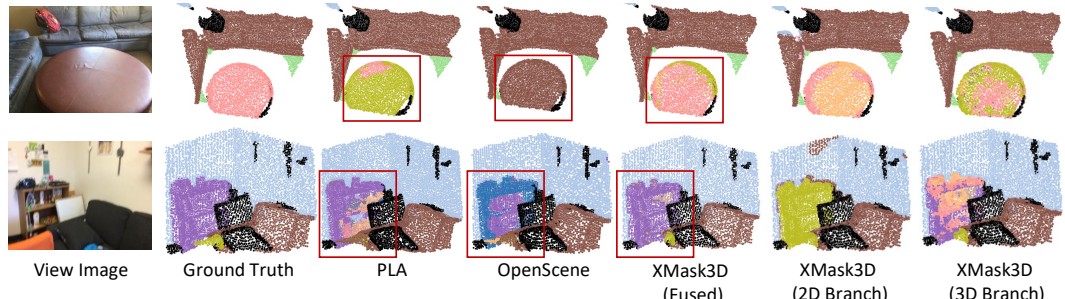

| View Image | Ground Truth | PLA | OpenScene | XMask3D (Fused) | XMask3D (2D Branch) | XMask3D (3D Branch) |

Figure 5: **Complete Visualization Comparisons.** We show comprehensive comparison between XMask3D fused/2D branch/3D branch outputs and previous methods (PLA [11]/OpenScene [34]). The figure corresponds to Figure 3 in the main paper.

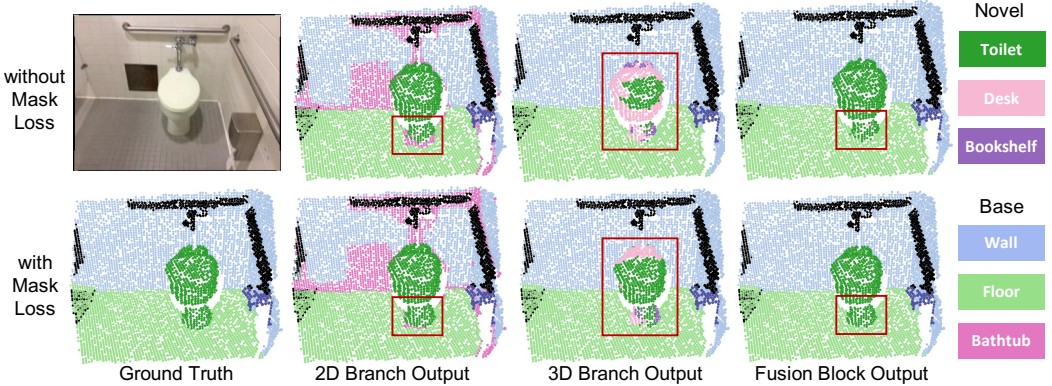

Figure 6: **Visualization Results of Ablations.** The first and second rows display segmentation results without and with the proposed mask regularization. The last three columns compare the outputs from the intermediate 2D and 3D branches with the final fusion block.

set $\omega_{\mathrm{mask}} = 0.5/0.5/1/2/2/1/1, \omega_{\mathrm{bi}} = 16/12/8/48/32/20/15$ for ScanNet B15/N4, B12/N7, B10/N9, ScanNet200 B170/N30, B150/N50, S3DIS B8/N4, B6/N6 benchmarks, respectively.

**Category Partitions.** We follow PLA [11] to partition the ScanNet [9] and S3DIS [1] datasets into several different benchmarks. Here we list the detailed partition principle in Table 4 for reference. We do not list splits for the ScanNet200 dataset since there are too many categories.

# B   Additional Experimental Results

## B.1   Per-class Results Comparison

We contrast per-class segmentation mean Intersection over Union (mIoU) with PLA [11] across the ScanNet and S3DIS datasets in Table 5. XMask3D significantly outperforms PLA, particularly in novel categories, thereby showcasing the effectiveness of our proposed cross-modal mask reasoning approach on open vocabulary 3D semantic segmentation.

## B.2   Visualization Comparison

In Figure 5, we show 2D and 3D branch outputs from XMask3D in addition to Figure 3 in the main paper. The 2D branch misclassifies the novel categories, while the 3D branch produces discontinuous segmentation masks. Neither of them outperforms PLA or OpenScene. However, when their features are fused together, the output becomes superior, demonstrating the complementary knowledge across different modalities.

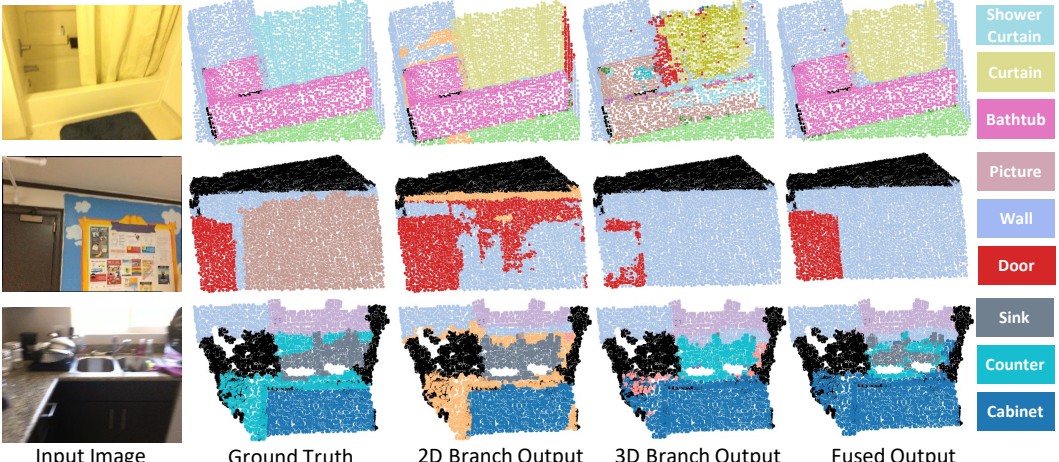

Figure 7: **Failure cases of XMask3D.** We focus on shower curtain, picture, and sink novel categories in each line, respectively.

### B.3 Illustrations of Ablations

We present supplementary illustrations of ablation studies pertaining to mask regularization and cross-modality fusion in Figure 6, which corresponds to the sample in Figure 1 from the ScanNet B15/N4 benchmark.

Comparing the first and second rows of the third column reveals that the proposed mask regularization loss yields significant improvements in the segmentation of novel categories by the 3D branch. Further comparison of the last three columns in the second row demonstrates that the output from the fusion block effectively leverages the strengths of both modality branches. Specifically, while the 2D branch excels at segmenting novel categories, it tends to produce discontinuous masks for base categories with larger regions (e.g., wall). Conversely, the 3D branch may perform relatively poorly in novel category segmentation but provides consistent geometric knowledge as a complementary aspect. Consequently, the fused output achieves superior performance in open vocabulary segmentation.

## C    Failure Cases Analysis

In Figure 7, we display some failure cases of XMask3D. The first sample shows a bathtub with a shower curtain in the bathroom. However, the 2D/3D branch and the fused output of XMask3D all misclassify the shower curtain as curtain. This may be because these two categories are similar in object shape and texture, with the only difference being the surrounding environment. Since XMask3D only takes a corner of the room as input instead of the entire scene, the global environmental information is insufficient for making the correct category prediction.

The second sample shows a large area of picture on the wall. The 2D, 3D branches and the fused output of XMask3D all misclassified it as wall, due to their similar geometry. In most cases, a picture is a small region on the wall, and this picture, as large as the wall, is a typical corner case. This failure case may reveal XMask3D's over-reliance on geometric knowledge and lesser consideration of texture information when encountering out-of-distribution samples.

The third sample shows a sink on a counter. Due to the occlusion problem, the sink point cloud is incomplete, negatively affecting the prediction of segmentation boundaries between the sink and the counter. This occurs because they are geometrically similar when the sinking-down part of the sink is missing.

