# OpenReview forum: "XMask3D: Cross-modal Mask Reasoning for Open Vocabulary 3D Semantic Segmentation"
_NeurIPS.cc/2024/Conference — NeurIPS 2024 poster_

### Official Review · Reviewer_tdkT · 2024-06-25

**Soundness:** 3
**Presentation:** 2
**Contribution:** 3
**Rating:** 5
**Confidence:** 3

**Summary:**

This paper introduces XMask3D, a framework developed for open vocabulary 3D semantic segmentation. They propose the integration of the denoising UNet, derived from a pre-trained diffusion model, to generate geometry-aware segmentation masks conditioned on learnable implicit 3D embeddings. These binary 2D masks are used to filter mask-level embeddings of 3D representations and apply mask regularization, thereby improving the open vocabulary capacity of 3D features.

**Strengths:**

1.	The motivation is clear.
2.	The proposed method is intuitive, and the experiments have validated their contributions.

**Weaknesses:**

1.	The organization should be improved. Section 3.1 provides an overview, while section 3.2 includes design insights and preliminary findings. The flow of these writings has puzzled me, making it difficult to grasp your key contribution.

**Questions:**

1.	Please provide further clarification on how mask-level alignment between 3D features and 2D embedding space can address the limitations of traditional techniques, such as global feature alignment or vision-language model distillation. Additionally, if texts (Category Labels) are concatenated with fused features, will it still create a unified feature space that encompasses 3D, 2D, and textual modalities?
2.	Could you please provide further clarification on the main contributions of your research compared to PLA and OpenScene? Although the 3D caption process shares similarities with PLA, the overall pipeline resembles OpenScene, with the exception of the diffusion model and mask generator, which differ from the Multi-view Feature Fusion in OpenScene.
3.	Does the Implicit 3D Captioner effectively work with your 3D features? From my understanding, the most reliable 3D captioner currently available is Cap3D, which generates captions for 3D objects by rendering multi-view images and utilizing BLIP2 and LLM for assistance. In the context of indoor-scenes, can we consider the Implicit 3D Captioner to be equally robust? It would be beneficial to present additional evidence to support this claim.
4.	Can your text-to-image diffusion model effectively generalize to your datasets? If not, please provide examples of failure cases. Additionally, is the diffusion model fine-tuned during the training process or is it frozen? If not, please present additional results to demonstrate the robustness of your diffusion model in generating high-quality images within your datasets.
5.	What is view-level contrastive loss? Why this loss is calculated between the view global feature and text embedding of the view image caption but have three coefficients?
6.	It is recommended to show your 2D Mask and 3D Mask in Figure 3 to provide more visual evidence.
7.	The authors should provide results on ScanNet++ (CVPR’23), which is a up-to-date dataset compared with ScanNet.
8.	Since diffusion models are utilized, it is recommended to compared the model parameters and FLOPs compared with PLA and OpenScene.

**Limitations:**

1.	The authors have addressed the limitations and potential negative societal impact of their work.

---

> ### Author Rebuttal · Authors · 2024-08-06
>
> Thanks for your careful review and constructive comments! Hopefully the following response will address your concerns.
>
> ### **1. Paper organization.**
>
> > The organization should be improved ...
>
> Thanks for your advice! We will reorganize the first two sections in our revised paper.
>
> ### **2. Clarifications.**
>
> #### **(A) Mask-level alignment.**
>
> > ... clarification on how mask-level alignment ...
>
> The most critical challenge in addressing the 3D open vocabulary problem is establishing 3D-text alignment. Previous approaches have tackled this by enforcing either global-wise or point-wise alignment between 3D features and 2D-text embeddings.
>
> Global feature alignment typically involves calculating a global feature of a scene or view through pooling and applying contrastive loss with 2D-text embeddings. However, pooling often result in the loss of detailed information, leading to unclear and imprecise boundaries on novel categories.
>
> Point-wise feature alignment matches 3D points with 2D pixels using camera parameters, and performs dense contrastive learning between them. While this method preserves boundary details, it incurs a significantly larger computational burden and less stable training, as the loss is highly susceptible to outliers.
>
> Our proposed mask-level alignment presents an intermediate solution of these two approaches. Compared to global features, mask features retain more detailed information since there are dozens of masks within a single view. Additionally, mask features are more robust than point features, as the distraction from outliers is mitigated by the mask pooling operation.
>
> > ... unified feature space ...?
>
> The 2D features with high open-vocabulary capacity are embedded in the 2D-text feature space. By applying 2D-to-3D mask regularization to the 3D features, these 3D features are drawn towards the 2D-text feature space. Consequently, the fused features, which integrate both 2D and 3D features, are expected to align well with the 2D-text embedding space. This alignment is further conveyed by the superior open-vocabulary performance of XMask3D compared to PLA and OpenScene. Thus, we believe that concatenating text and fused features will effectively create a unified feature space that integrates 3D, 2D, and textual modalities.
>
> #### **(B) Main contributions compared with OpenScene and PLA.**
>
> > ... main contributions compared to PLA and OpenScene ...
>
> Sorry for not directly highlighting our contributions relative to OpenScene and PLA.
>
> OpenScene employs point-wise distillation and feature ensemble, while XMask3D utilizes mask-wise feature regularization. As discussed in *Section 2(A)*, mask-wise contrastive learning is more robust and less computationally demanding compared to point-wise counterpart.
>
> PLA proposes entity-level point-caption association through set difference and intersection operations, which is less precise and adaptable than our mask-level 3D-2D-text association. The mask for 3D-text alignment is adaptively predicted by the 2D mask generator, making XMask3D an end-to-end and integrated system.
>
> Therefore, the core contribution of our paper is the introduction of a more adaptive and robust mask-level alignment technique.
>
> #### **(C) Implicit 3D captioner.**
>
> > Does the Implicit 3D Captioner ...
>
> According to the ablation studies in Table 3(a), the proposed implicit 3D captioner yields better novel class results compared to using vanilla text embeddings or implicit 2D caption embeddings, which convey the effectiveness of the implicit 3D captioner. However, unlike Cap3D, the implicit 3D captioner does not directly generate text captions. It only produces conditional features that work effectively with the XMask3D pipeline and we cannot guarantee its robustness in other 3D-to-text generation scenarios.
>
> #### **(D) Text-to-image diffusion model.**
>
> > Can your text-to-image diffusion model ...
>
> In our XMask3D pipeline, all weights of the denoising UNet are frozen. We do not generate images using the text-to-image diffusion model. Instead, we use the HxWxC features from the denoising UNet to generate 2D masks through the mask generator. Only the mask generator is trained on our datasets.
>
> #### **(E) View-level contrastive loss.**
>
> > What is view-level contrastive ...
>
> Sorry for the unclear statement. The ground truth for the view-level contrastive loss is the text embedding of the view image caption. The predictions are derived from the average pooling results of the 3D branch features, 2D branch features and fused features respectively. Therefore, we have three view-level contrastive losses and design three separate coefficients for them.
>
> ### **3. Additional visualizations.**
>
> > ... 2D Mask and 3D Mask in Figure 3 ...
>
> Thanks for your insightful advice! In Figure 4 in the PDF attachment of the global response, we display the outputs from the 2D and 3D branches for the same samples shown in Figure 3 of our main paper.
>
> ### **4. ScanNet++ dataset.**
>
> > ... results on ScanNet++ ...
>
> We apologize for not providing results on ScanNet++, as we followed the protocols of previous papers (OpenScene and PLA). Following your advice, we have submitted a request to download ScanNet++, but unfortunately it is still pending, so we are unable to conduct experiments within the rebuttal period. We appreciate your suggestion regarding this advanced dataset and plan to add it to our future work.
>
> ### **5. Resource comparisons.**
>
> > ... model parameters and FLOPs ...
>
> Thanks for your suggestion! As the FLOPs of 3D models are conditioned on point cloud numbers, we only report the additional FLOPs of the 2D branch. In future work, we plan to replace the 2D branch with a more efficient and lightweight open vocabulary 2D mask generator.
>
> |Method|Trainable Params|Non-trainable Params|Extra FLOPs|
> |:-:|:-:|:-:|:-:|
> |PLA|11.0 M|--|--|
> |OpenScene|15.6 M|126.5 M|OpenSeg: 1.6 TFLOPs|
> |XMask3D|82.9 M|1493.8 M|Denoising UNet: 4.3 TFLOPs|
> ||||Mask Generator: 93.5 GFLOPs|

---

> > ### Comment · Reviewer_tdkT · 2024-08-10
> > **Reviewer response**
> >
> > Thanks for your responses, most of my concerns have been solved. I will change my rating to positive. Please carefully release your code and checkpoints in your future version.

---

> > > ### Author Response · Authors · 2024-08-12
> > > **Response to Reviewer tdkT**
> > >
> > > Thanks for upgrading your score and providing valuable feedback. We will update our revised paper according to our discussions and release our code and checkpoints after the conference decision. Thank you again for your insightful and constructive suggestions that improve paper quality!

---

### Official Review · Reviewer_AnYT · 2024-07-12

**Soundness:** 3
**Presentation:** 3
**Contribution:** 3
**Rating:** 5
**Confidence:** 4

**Summary:**

The paper proposes a precise and consistent mask-level alignment between 3D features and the 2D-text embedding space through a method called cross-modal mask reasoning. The proposed XMask3D model includes a 3D branch for capturing geometric features, a 2D branch for generating vision-language aligned masks, and a fusion block to combine 3D with 2D. Using a pre-trained text-to-image diffusion model as the 2D mask generator, the model leverages three techniques: 3D-to-2D mask generation, 2D-to-3D mask regularization, and 3D-2D mask feature fusion.

**Strengths:**

1- The idea is novel, the author propose to merge 2D which provides high OV capabilities, with 3D features shich endoces 3D geometry.

2- The method performs remarkably better than the reported models, namely OpenScene. The experiments are also well structure

**Weaknesses:**

1- The authors don't compare with state-of-the-art 3D semantic segmentation OV3D[1]

2- The authors highlighed fututre work in the limitation, it would be good if you can expand it with some limitation on the technical side or some failure cases.

[1] Jiang, Li, Shaoshuai Shi, and Bernt Schiele. "Open-Vocabulary 3D Semantic Segmentation with Foundation Models." Proceedings of the IEEE/CVF Conference on Computer Vision and Pattern Recognition. 2024.

**Questions:**

Please compare to OV3D mentioned in the weaknesses

**Limitations:**

Needs to be expanded

---

> ### Author Rebuttal · Authors · 2024-08-06
>
> Thanks for your careful review and constructive comments! Hopefully the following response will address your concerns.
>
> ### **1. Results comparison.**
>
> > The authors don't compare with state-of-the-art 3D semantic segmentation OV3D.
>
> Thanks for your suggestion! We will include this outstanding method in the ScanNet comparison in our revised paper. OV3D primarily focuses on EntityText extraction and Point-EntityText association, while XMask3D concentrates on enhancing mask-level interaction between 2D and 3D modalities. Therefore, the contributions of OV3D and XMask3D are orthogonal and could complement each other. Once the official code of OV3D is released, we plan to integrate our model with OV3D's advanced design in prompting LLM-powered LVLM models and Pixel-EntityText alignment to further enhance the performance of XMask3D.
>
> ### **2. Expanded limitations.**
>
> > The authors highlighed fututre work in the limitation, it would be good if you can expand it with some limitation on the technical side or some failure cases.
>
> #### **(A) Technical limitations.**
>
> Currently, the denoising UNet from the pre-trained diffusion model requires significant computational resources and impacts inference efficiency. In contrast to PLA, which consists solely of a 3D model, XMask3D could be further improved by replacing the 2D branch with a more lightweight and efficient 2D open vocabulary mask generator.
>
> #### **(B) Failure case analysis.**
>
> Thanks for your constructive suggestion! We provide three failure cases of XMask3D in Figure 1 in the PDF attachment of the global response. We will include this figure and the following discussions in the supplemental material of our revised paper.
>
> The first sample shows a `bathtub` with a `shower curtain` in the bathroom. However, the 2D/3D branch and the fused output of XMask3D all misclassify the `shower curtain` as `curtain`. This may be because these two categories are similar in object shape and texture, with the only difference being the surrounding environment. Since XMask3D only takes a corner of the room as input instead of the entire scene, the global environmental information is insufficient for making the correct category prediction.
>
> The second sample shows a large area of `picture` on the `wall`. The 2D/3D branch and the fused output of XMask3D all misclassified it as `wall`, due to their similar geometry. In most cases, a picture is a small region on the wall, and this picture, as large as the wall, is a typical corner case. This failure case may reveal XMask3D's over-reliance on geometric knowledge and lesser consideration of texture information when encountering out-of-distribution samples.
>
> The third sample shows a `sink` on a `counter`. Due to the occlusion problem, the `sink` point cloud is incomplete, negatively affecting the prediction of segmentation boundaries between the `sink` and the `counter`. This occurs because they are geometrically similar when the sinking-down part of the sink is missing.

---

> > ### Comment · Reviewer_AnYT · 2024-08-12
> >
> > Thanks a lot for clarifying these points. The authors addressed my comments. Thus, I am happy to keep my score, and it would be great to report the results of OV3D and XMask3D together in the revised version.

---

> > > ### Author Response · Authors · 2024-08-13
> > > **Response to Reviewer AnYT**
> > >
> > > We deeply appreciate the time and effort you dedicated to the careful review and insightful feedback on our paper! We will update our revised paper and add OV3D comparison according to our discussions.

---

### Official Review · Reviewer_Xo8p · 2024-07-12

**Soundness:** 3
**Presentation:** 3
**Contribution:** 3
**Rating:** 7
**Confidence:** 4

**Summary:**

The paper addresses the limitations of current open vocabulary 3D semantic segmentation methods, which primarily focus on creating a unified feature space for 3D, 2D, and textual modalities but struggle with fine-grained segmentation boundaries. To overcome these limitations, the authors propose XMask3D, a cross-modal mask reasoning framework that achieves more precise mask-level alignment between 3D features and the 2D-text embedding space.

**Strengths:**

1. The part "incorporating a 2D mask generator to create geometry-aware open masks and apply fine-grained mask-level regularization on 3D features" seems reasonable and novel.
2. The paper is well-structured and easy to follow.
3. Analysis is thorough and insightful.

**Weaknesses:**

1. The paper evaluates the proposed method on a limited set of benchmarks (ScanNet20, ScanNet200, S3DIS), all of which are indoor scene datasets. Authors could discuss how the method might perform on outdoor datasets. Additionally, the authors could provide a qualitative analysis of the model's potential limitations when applied to different environments.
2. The reliance on the denoising UNet from a pre-trained diffusion model could be seen as a potential weakness or limitation, especially given the computational resources required for training and inference.

**Questions:**

1. The paper could benefit from a more detailed error analysis to understand the failure modes of XMask3D, especially in novel category segmentation.

**Limitations:**

The authors have addressed limitations.

---

> ### Author Rebuttal · Authors · 2024-08-06
>
> Thanks for your careful review and constructive comments! Hopefully the following response will address your concerns.
>
> ### **1. Application of XMask3D on other scenarios.**
>
> > The paper evaluates the proposed method on a limited set of benchmarks (ScanNet20, ScanNet200, S3DIS), all of which are indoor scene datasets. Authors could discuss how the method might perform on outdoor datasets.
>
> We follow the experimental settings outlined in PLA for our experiments on indoor scene datasets. Based on your advice, we observed that other open vocabulary papers [1,2] also conduct experiments on the outdoor scene dataset nuScenes[3]. However, this dataset only provides point cloud data with annotations and raw image data without annotations. Without image-level segmentation ground truth, we cannot train the 2D mask generator. In our future work, we plan to replace the entire 2D branch with a pre-trained 2D open vocabulary segmentation model, enabling us to handle the nuScenes dataset without requiring 2D annotations.
>
> However, we have made efforts to provide some qualitative results on the nuScenes dataset. First, we project the 3D annotations onto 2D images using the camera's intrinsic and extrinsic parameters. Due to the sparsity of the 3D point cloud, we further employ the k-nearest-neighbor algorithm to fill in the blank regions without pixel-point correspondence, as shown in Figure 3 in the PDF attachment of the global response. The projected 2D label maps and the outputs from XMask3D on several nuScenes samples are shown in Figure 2, demonstrating the potential of XMask3D in handling outdoor scenarios.
>
> > Additionally, the authors could provide a qualitative analysis of the model's potential limitations when applied to different environments.
>
> As discussed above, the current architecture of XMask3D requires full annotations for both 2D and 3D data. Therefore, when dealing with outdoor datasets that lack fine-grained 2D annotations, the performance of XMask3D may decrease. Additionally, we provide a qualitative analysis of failure cases in `3. Failure case analysis`, which reveals potential limitations of XMask3D under various circumstances.
>
> ### **2. Computational costs.**
>
> > The reliance on the denoising UNet from a pre-trained diffusion model could be seen as a potential weakness or limitation, especially given the computational resources required for training and inference.
>
> We acknowledge that the denoising UNet requires significant computational resources and slows down the inference speed. In our future work, we plan to address this limitation by replacing the 2D branch with a more lightweight 2D open vocabulary mask generator. Thank you for highlighting this potential weakness, which has inspired our future improvements.
>
> ### **3. Failure case analysis.**
>
> > The paper could benefit from a more detailed error analysis to understand the failure modes of XMask3D, especially in novel category segmentation.
>
> Thanks for your constructive suggestion! We provide three failure cases of XMask3D in Figure 1 in the PDF attachment of the global response. We will include this figure and the following discussions in the supplemental material of our revised paper.
>
> The first sample shows a `bathtub` with a `shower curtain` in the bathroom. However, the 2D/3D branch and the fused output of XMask3D all misclassify the `shower curtain` as `curtain`. This may be because these two categories are similar in object shape and texture, with the only difference being the surrounding environment. Since XMask3D only takes a corner of the room as input instead of the entire scene, the global environmental information is insufficient for making the correct category prediction.
>
> The second sample shows a large area of `picture` on the `wall`. The 2D/3D branch and the fused output of XMask3D all misclassified it as `wall`, due to their similar geometry. In most cases, a picture is a small region on the wall, and this picture, as large as the wall, is a typical corner case. This failure case may reveal XMask3D's over-reliance on geometric knowledge and lesser consideration of texture information when encountering out-of-distribution samples.
>
> The third sample shows a `sink` on a `counter`. Due to the occlusion problem, the `sink` point cloud is incomplete, negatively affecting the prediction of segmentation boundaries between the `sink` and the `counter`. This occurs because they are geometrically similar when the sinking-down part of the sink is missing.
>
> ### **References**
> [1] Jihan Yang, et al. "RegionPLC: Regional Point-Language Contrastive Learning for Open-World 3D Scene Understanding." CVPR. 2024.
> [2] Qingdong, He, et al. "UniM-OV3D: Uni-Modality Open-Vocabulary 3D Scene Understanding with Fine-Grained Feature Representation." IJCAI. 2022.
> [3] Holger Caesar, et al. "nuscenes: A multimodal dataset for autonomous driving." CVPR. 2020.

---

> > ### Comment · Reviewer_Xo8p · 2024-08-08
> >
> > I've thoroughly reviewed the authors' responses and appreciate their thoughtful engagement. Most of my concerns have been addressed. I will stay in touch for further discussion as we approach the final rating.

---

> > > ### Author Response · Authors · 2024-08-12
> > > **Response to Reviewer Xo8p**
> > >
> > > We greatly appreciate your response and valuable suggestions, which improved the quality and comprehensiveness of our paper! We will update our revised paper according to our discussions.

---

### Official Review · Reviewer_4VWC · 2024-07-14

**Soundness:** 3
**Presentation:** 3
**Contribution:** 3
**Rating:** 6
**Confidence:** 4

**Summary:**

This paper addresses the challenge of open-vocabulary 3D semantic segmentation by utilizing 3D geometric features, 2D semantic embeddings, and text modality. The proposed approach adapts the ODISE method to the 3D domain, aiming to distill open-vocabulary semantic segmentation knowledge from a pre-trained text-to-image denoising diffusion model to a 3D segmentation model. Initially, an input point cloud is fed into a 3D encoder-decoder segmentation network, producing point-wise geometric features. Simultaneously, a pre-trained visual-language diffusion model generates 2D masks and embeddings from posed images of the same scene, conditioned on the 3D global feature of the 3D branch’s encoder. Unlike the ODISE method, an implicit $3D$ captioner is introduced to produce geometry-aware 2D masks while also distilling information from the 2D branch network to the 3D encoder. To further regularize the 3D network, a distillation loss ($\mathcal{L}_{mask}$) is applied to the 3D mask embeddings, derived from the per-point features and the 2D masks back-projected to the point cloud as 3D binary masks. By obtaining ground truth mask features from a pre-trained CLIP model, the 3D masked embeddings are aligned with the image-text joint embedding space, through a cosine similarity loss. This alignment leads to more coherent segmentation results and enhances the model's open-vocabulary capabilities. Finally, the per-point features are combined with the pseudo mask 2D features (formed by the back-projected 3D mask and 2D mask embeddings), resulting in a fused per-point representation that incorporates the geometric information from the 3D segmentation network and the semantic open-vocabulary capabilities of the 2D branch. The approach is evaluated on three semantic segmentation benchmarks (ScanNet, ScanNet200, and S3DIS) and demonstrates superior performance compared to competing methods.

**Strengths:**

The XMask3D effectively aligns 3D geometric features with 2D and textual modailities through knowledge distillation from visual-text joint embedding spaces inherent in the pre-trained 2D denoising UNet and the CLIP model. As evident by the ablation, the implicit 3D captioner is a crucial step in the overall pipeline, and it outperforms vanilla text conditioning or the implicit 2D captioner of ODISE, in both base and novel semantic categories. Moreover, the 2D-to-3D mask regularization is also essential, since it significantly improves the accuracy of the proposed method esp. in novel categories. This justifies the need for this additional distillation step from the CLIP joint space, to further enhance the open-vocabulary capabilities of the XMask3D method. Finally, the discussion on modality fusion, both in the main paper and supplementary, is highly appreciated. By dissecting the method and providing qualitative and quantitative results for each step, the authors make it easier for readers to understand and gain intuition about the presented approach.

**Weaknesses:**

While the method exhibits superior performance w.r.t. competing methods, it seems that the output fused embeddings yields to geometric inconsistent features for semantic classes that cover large areas of the point cloud such as wall, ceiling and floor. This is evident in both partitioning settings when the class is either base or novel (Table 5 (a) and (b) in supp.).

**Questions:**

Following the weaknesses section, do the authors have any additional insights into why this phenomenon occurs with the fused embeddings? Could the 2D regularization terms ($\mathcal{L}_{seg}^{2D}$, $\mathcal{L}\_{view}^{2D}$) be introducing too much bias towards the 2D visual modality, thereby causing geometric discontinuities in the output fused features?

**Limitations:**

Yes, the authors have discussed the method's limitations in detail in Section 4.4.

---

> ### Author Rebuttal · Authors · 2024-08-06
>
> Thanks for your careful review and constructive comments! Hopefully the following response will address your concerns.
>
> ### **1. About the less satisfactory results of classes that cover large areas.**
> > While the method exhibits superior performance w.r.t. competing methods, it seems that the output fused embeddings yields to geometric inconsistent features for semantic classes that cover large areas of the point cloud such as wall, ceiling and floor. This is evident in both partitioning settings when the class is either base or novel (Table 5 (a) and (b) in supp.).
>
> The underlying reasons for the unsatisfactory performances of semantic classes on the ScanNet and S3DIS datasets are distinct. For the ScanNet dataset, XMask3D processes only a corner of the scene in each forward pass, whereas PLA directly feeds the entire scene into the model. This segmentation approach negatively impacts categories that cover large areas, such as `wall`, `ceiling`, and `floor`, because cutting these categories into smaller pieces inevitably undermines segmentation performance by losing long-range relational information.
>
> For the S3DIS dataset, in addition to the aforementioned partial input issue, we encounter a trade-off between computational resource consumption and segmentation performance. The validation samples in S3DIS are densely populated, with an image view potentially corresponding to over 500,000 points, which causes out-of-memory errors on a 24GB NVIDIA 4090 device and slow inference speed. Consequently, we discard views exceeding a threshold of 260,000 points for higher efficiency. However, this view selection process results in hollow regions in the merged outputs, primarily in the corners of rooms, and these regions typically correspond to the categories `wall`, `ceiling`, or `floor`. We conducted an ablation study, increasing the selection threshold from 260,000 to 500,000 points, as shown in the following table. When the threshold is raised, the segmentation performance for `ceiling`, `floor`, and `wall` consistently improves.
>
> | Partition | Threshold | Ceiling | Floor | Wall |
> | :-------: | :-------: | :-----: | :---: | :--: |
> | B8/N4     | 260,000   | 86.4    | 88.3  | 81.4 |
> |           | 500,000   | 90.1    | 91.1  | 81.6 |
> | B6/N6     | 260,000   | 86.4    | 47.4  | 80.9 |
> |           | 500,000   | 88.7    | 51.1  | 81.6 |
>
> > Following the weaknesses section, do the authors have any additional insights into why this phenomenon occurs with the fused embeddings? Could the 2D regularization terms ($L_{seg}^{2D}$, $L_{view}^{2D}$) be introducing too much bias towards the 2D visual modality, thereby causing geometric discontinuities in the output fused features?
>
> As discussed above, the primary reasons are the partial point cloud input and the trade-off with memory consumption. We believe that the 2D regularization term does not significantly bias the fused features towards geometric discontinuities. As illustrated in Figures 1, 4, and 6 of our main paper, the fused features of `wall` and `floor` fully leverage the continuous geometry from the 3D branch. Additionally, we provide a qualitative analysis in the following table. The 2D branch performance is significantly worse than the others. The fused features perform better than the 3D features for `wall` and only slightly worse for `floor`. The numerical results also demonstrate that the performance of fused features in segmenting `wall` and `floor` is not significantly affected by geometric discontinuities from the 2D regularization terms.
>
> | Partition | Method  | Branch | Wall | Floor |
> | :-------: | :-----: | :----: | :--: | :---: |
> | B10/N9    | PLA     | 3D     | 84.6 | 95.0  |
> |           | XMask3D | Fused  | 84.2 | 94.7  |
> |           | XMask3D | 3D     | 82.0 | 95.1  |
> |           | XMask3D | 2D     | 62.9 | 78.6  |
> | B12/N7    | PLA     | 3D     | 84.7 | 95.1  |
> |           | XMask3D | Fused  | 83.3 | 94.6  |
> |           | XMask3D | 3D     | 81.8 | 95.0  |
> |           | XMask3D | 2D     | 60.3 | 76.7  |
> | B10/N9    | PLA     | 3D     | 83.8 | 95.2  |
> |           | XMask3D | Fused  | 83.8 | 94.7  |
> |           | XMask3D | 3D     | 81.7 | 95.1  |
> |           | XMask3D | 2D     | 64.3 | 78.5  |

---

> > ### Comment · Reviewer_4VWC · 2024-08-14
> >
> > Thank you for the clarifications and for addressing my concerns. I will maintain my positive rating

---

> > > ### Author Response · Authors · 2024-08-14
> > > **Response to Reviewer 4VWC**
> > >
> > > We greatly appreciate your response and valuable suggestions, which improved the quality and comprehensiveness of our paper! We will update our revised paper according to our discussions.

---

### Author Rebuttal · Authors · 2024-08-06

We would like to thank all the reviewers for taking the time to review our submission and provide constructive feedback on our work. We are encouraged by the consensus among reviewers regarding the strengths of our approach, which aligns with our intentions and efforts:

1. **Novelty and Significance**: We appreciate that all reviewers recognize our approach as both novel and reasonable. It is encouraging to see comments such as those from Reviewer AnYT, who noted that "The idea is novel," and Reviewer Xo8p, who remarked that "the part 'incorporating a 2D mask generator to create geometry-aware open masks and apply fine-grained mask-level regularization on 3D features' seems reasonable and novel."
2. **Thorough Evaluation**: We are pleased that the reviewers acknowledged the comprehensiveness of our experiments and evaluations. Reviewer Xo8p commented that "Analysis is thorough and insightful," while Reviewer tdkT noted that "The proposed method is intuitive, and the experiments have validated their contributions."
3. **Clarity and Presentation**: We are gratified that our efforts to present our ideas clearly have been well-received. Reviewer 4VWC highlighted that "By dissecting the method and providing qualitative and quantitative results for each step, the authors make it easier for readers to understand and gain intuition about the presented approach." Reviewer Xo8p also mentioned that "The paper is well-structured and easy to follow."

We also appreciate the suggestions for further improving our work. In response to the specific concerns and recommendations raised by each reviewer, we have provided detailed discussions in our rebuttal and will update them in our revised paper. Additionally, we have prepared a PDF with additional illustrations to offer a more comprehensive visualization of our results and reinforce the validity of our work.

Best regards,

Submission 637 Authors

---

### Comment · Area_Chair_Eak8 · 2024-08-10
**Discussions and finalize rating**

Dear Reviewers,

Thank you for your efforts. Please review the rebuttals, engage in the discussions, and provide your final ratings.

Thank you again for your valuable contributions.

AC

---

### Decision · Program_Chairs · 2024-09-25

**Decision:**

Accept (poster)

**Comment:**

This paper introduces XMask3D, an approach to open-vocabulary 3D semantic segmentation that addresses key limitations in existing methods by integrating 3D geometric features with 2D and text-based embeddings. The authors build on the ODISE method, extending it to the 3D domain and introducing a novel implicit captioner that generates geometry-aware 2D masks. This enhances the alignment between 3D and 2D-text embedding spaces, resulting in improved segmentation accuracy, particularly for novel categories. The method employs a design incorporating distillation losses and mask-level regularization, which boosts its open-vocabulary capabilities.

XMask3D is evaluated on several benchmarks, including ScanNet, ScanNet200, and S3DIS, where it demonstrates superior performance compared to existing methods. These results are further supported by a comprehensive ablation study. All reviewers have recommended acceptance of this paper, and the AC finds no significant reasons to overturn these recommendations. Therefore, the paper is recommended for acceptance.